# Evaluation of In-Service Vocational Teacher Training Program: A Blend of Face-to-Face, Online and Offline Learning Approaches

Muhammad Zaheer Asghar [1,2,*,†], Muhammad Naeem Afzaal [2,†], Javed Iqbal [2,3], Yasira Waqar [4] and Pirita Seitamaa-Hakkarainen [5]

1  Crafts Sciences Unit, Department of Teacher Education, University of Helsinki, 00014 Helsinki, Finland
2  Department of Education, University of Management and Technology, Lahore 54770, Pakistan
3  Faculty of Management Science, Ilma University Karachi, Karachi 75190, Pakistan
4  School of Education, Lahore University of Management Sciences, Lahore 54792, Pakistan
5  Department of Education, University of Helsinki, 00014 Helsinki, Finland
*  Correspondence: zaheer.asghar@helsinki.fi
†  These authors contributed equally to this work.

**Abstract:** Teacher education has a large and favorable impact on a teacher's performance. Effective training brings improvement in the productivity and performance of employees due to the acquisition of new knowledge and skills. The National Vocational and Technical Training Commission organized an in-service vocational teachers' training program through blended learning approaches in Pakistan. This study aimed at finding out the effectiveness of that in-service vocational teachers' training program. The four levels of the Kirkpatrick model were used as the theoretical framework. A survey approach was used to collect data from 629 in-service vocational teachers who took part in the training program through blended learning approaches. Partial least square structural equation modeling was applied to find the hierarchal relationship among the four levels of the Kirkpatrick model. The results of the current research revealed that trainees were satisfied at all four levels of the Kirkpatrick model, including the reaction, learning, behavior and results. A hierarchal relationship between the four levels of the Kirkpatrick model was also found for the evaluation of the training program. The direct effect of the reaction had a positive and significant relationship with learning, learning with behavior and behavior with the results. The results of the specific indirect relationship among the four levels clearly depicted that behavior was mediated through learning and the results, learning was mediated through the reaction and behavior and behavior was mediated through the reaction and results. This study is useful for vocational education institutions and skill development policymakers to design in-service training programs for in-service teachers. Future studies can be conducted about the adaptation of instructions for the in-service vocational teachers' training program through blended learning approaches.

**Keywords:** blended learning; Kirkpatrick model; vocational education; in-service teachers

## 1. Introduction

Education that prepares learners for skill development and employment in various economic fields is referred to as career, technical and vocational training [1]. The role of technical and vocational teachers is crucial in this regard and, therefore, emphasis must be placed on their development in both professional and pedagogical aspects. Teacher training improves the teaching practices of teachers, which results in the better learning of their students [2]. The majority of school reforms and strategies place emphasis on professional development and teacher quality [3–5]. The National Vocational and Technical Training Commission (NAVTTC) in Pakistan is the apex body at a federal level that is responsible for the development of technical and vocational education. Its core objective is to formulate policy to guide and direct the TVET sector in Pakistan and to also provide solutions to their challenges. The National Skills Strategy 2009–2013 (NSS, 2009) is a document

formulated by the NAVTTC to provide policy directions with the necessary support for implementation to achieve the national goals of skill development, so that improved social and economic conditions could be achieved (NSS, 2009). According to the NAVTTC [6], the TVET system of Pakistan is facing many challenges in the form of less trained people being of sub-standard quality; thus, being unable to compete in the national and international market and, consequently, a smaller number of people obtain the proper jobs relevant to their occupation. The NAVTTC [6] came up with various solutions to provide quality education in the field of TVET. For this purpose, several actions were proposed, out of which the training of in-service vocational teachers was of utmost importance. According to the authors of [7], teacher training is the most effective and interactive way to provide a guarantee of the successful development of teachers, whether it happens face-to-face [8–10] or online [11].

Technology advancements in recent decades have forced many organizations to make their employees more productive to compete in a rapidly changing world [12]. Fuller et al. [13] argued that organizations provide their employees with opportunities for training to improve their knowledge, skills and attitude relevant to their job role. Consequently, the NAVTTC, in coordination with GIZ, launched a training program for in-service vocational teachers in the field of pedagogy that adopted an innovative approach of blended learning. Instructional staff who are committed to providing the best education to their students always try to practice new teaching strategies and tools [14]. The blended learning approaches were adopted for the very first time in the history of Pakistan to train TVET teachers.

According to Nortvig et al. [15], blended learning has been viewed differently, and its definition changes with time; there were far fewer discussions over the terms "online" and "face-to-face" learning. When the technology was not involved in education, blended learning was thought to be the use of different teaching strategies in the classroom to facilitate learning [16]. When face-to-face teaching was linked with technology, such as e-learning and online learning for facilitating student learning, Nortvig, Petersen and Balle [15] noted that the focus of blended learning then shifted to the use of a mixed learning environment, i.e., face-to-face and online [16]. The definition, which is generally acceptable, is the combination of online, face-to-face and offline learning in the same training course [17]. The blended learning format used in Pakistan to train TVET teachers also included online, offline and face-to-face learning. It was completed in two phases: in the first phase, 100 master trainers were trained in 2012 and 2013, and in the second phase, these master trainers further imparted training to 8500 TVET teachers across Pakistan. In-service vocational teachers were from different subject areas, such as computing, electrical, dressmaking, paramedics, etc.

There is much less available empirical research on teachers' professional development with the blended learning approach [18]. Previous research covered blended learning in the general education sector and measured e-learning content, student achievements and the application of the innovative approach. For example, a piece of research was conducted to investigate the use of android-based learning materials for junior school classes [19]. Another piece of research was also conducted on science teachers at a secondary school with a blended learning approach to ascertain their ICT knowledge [20]. Student perceptions were measured about the application of the blended learning in previous research on university students [21]. The relationship of blended learning with student achievements was also measured on primary school students [22]. Thus, previous research showed that the effectiveness of the blended learning approach was not yet measured in the vocational education sector. The present study aimed to evaluate the training effectiveness of the blended learning approach adopted for in-service vocational education teacher training in Punjab.

Different approaches have been applied to evaluate different programs and projects. Approximately 22 approaches were gathered into five categories by Stufflebeam and Shinkfield [23]. These were categorized as pseudo-evaluations, quasi-evaluation studies, decision/accountability-oriented studies, improvement- and accountability-oriented evalu-

ations and social agenda/advocacy-directed approaches. The various sub-categories were also available for the procedures as mentioned earlier. Based on different approaches, different models for evaluation were presented by various researchers, such as Kirkpatrick's, the CIPP, the experimental/quasi-experimental, the logic and Robert Stake's countenance model [24]. According to Royse et al. [25], no model is of less importance, but they can be best used as per the needs and requirements of researchers. Kirkpatrick's model was formulated, widely reviewed and applied in social sciences for evaluating programs and their effectiveness [26]. The Kirkpatrick model was designed for objectively measuring training effectiveness. Each level represents a more precise measure of the effectiveness of a training program. Other models cover only one or two aspects of a training program, such as the CIPP model, which is useful for providing information during program development. In contrast, the Kirkpatrick model is applied after a program is implemented. It also covers the evaluation of online learning. The blended learning approach covers online learning, so the Kirkpatrick evaluation model fits well to evaluate the training program in totality. The research question under study was "what is effectiveness of in-service vocational teachers' training program through blended learning approaches"?

This study could be used as additional knowledge to previous research for finding out the effectiveness of in-service vocational education teacher training programs through blended learning approaches in Punjab, Pakistan. There was a gap in the research study of teacher training with blended learning approaches in the vocational sector, so our paper could provide guideline to teacher training providers to plan their professional development training program as per the recommendations of the current study. The present research provides knowledge about the provision of in-service TVET teacher training programs for TVET institutions in Pakistan, as well as the effectiveness of in-service TVET teacher training programs for the vocational institutes of Pakistan. There are less research about the provision of in-service TVET training programs through blended learning. The additional knowledge to previous existing investigations was distinguished through the application of a theoretical framework for evaluating in-service TVET teacher training programs. Moreover, two levels (reaction and learning) of the Kirkpatrick model have been used in previous research in different fields of education [27]. This research is a new addition in the application of the four levels of the Kirkpatrick model in the field of TVET teacher training.

This study focuses on exploring the effectiveness of existing training programs of TVET. This study provides guidelines for policymakers, including political and TVET leadership related to the existing skill development of Pakistan policies. Moreover, this study also provides some valuable suggestions to donor agencies. This study could potentially allow for the NAVTTC to reflect on their provision of in-service TVET teacher training through blended learning approaches. It could also provide an opportunity to review the existing training programs of TVET. This research can be a beacon of hope for policymakers to give them an insight into the execution of the existing skill development of Pakistan policies, and prove to be an invaluable source to politicians for formulating TVET policies in a better way in the future. Finally, it could also be a source of awareness for donor agencies about the utilization of their aids in the field of TVET, so that they can channelize their aids in a better way. The above points are discussed, providing the insight to understand how existing TVET training programs work and how they can be improved upon.

## 2. Theoretical Framework

Blended learning is a kind of teaching model that encompasses both face-to-face and online learning, which means students receive real time and independent engagement. Real-time student engagement means synchronous environments, and independent student engagement means asynchronous environments [28–30]. Blended learning, which combines online and face-to-face learning, has been touted as having numerous benefits [31,32], since the various modes, i.e., online, offline and face-to-face, can complement and mutually reinforce one another [33,34]. Technology plays a key role in nurturing the

blended learning approach [35]. It also improves school accountability, teaching skills, communication skills, creation and the analytical approach [36,37]. According to Alvarado-Alcantar, Keeley and Sherrow [28], enrolment at the secondary level of the blended learning approach increased 500% from 2002 to 2010 in America, which shows the acceptability of the blended learning approach.

This is the age of information technology, and many institutions have started to impart online learning [38,39]. Due to drastic changes in technology, students have multiple options of how to interact with teachers and their class fellows. This is in practice in the education system, but in technical and vocational education, online learning has not been fully adopted. Therefore, it was the need of the hour to develop a teacher training program in the field mentioned above. Online learning provides solutions to problems faced in conventional education; however, it also imposes some problems, e.g., separation, remoteness, inadequate feedback, distancing from group members and a lack of professionalism [40–42]. In this critical situation, blended learning emerged as a solution to problems faced in both approaches, as it meets online and teaching needs [43]. Teachers and students are more satisfied with the blended learning approach as compared to the single approach due to its effectiveness. It provides the facility of using different resources that improve the interaction between students and teachers and self-paced learning [43].

Moore, Robinson, Sheffield and Phillips [37] argued that higher education institutions are targeted to develop their faculty by adopting the blended learning approach as presented in previous research. There is a lot less research available showing the implementation of blended learning in the professional development of teachers at the secondary level, and only a few studies with guidance on how the blended learning approach affects instructional design and planning [37]. Different blended learning frameworks and models are available that provide standards to assess teachers' integration of blended learning [44], but research is still required for the evaluation of the in-service teacher training of vocational teachers through the use of the blended learning approach [45]. According to Parks et al. [46], the requirement of teachers for any kind of training is very important, and should be considered while planning professional development courses. It was further suggested that the blended program should be research-oriented, relevant and must have a long-term impact. According to Shand and Farrelly [47], a training program with a combination of pedagogy and technology results in effective learning. Ref. [48] argued that the best practices of a conventional training program should be adopted along with online considerations in a blended learning program. The best way is the alignment of online standards, best teaching practices of conventional teaching and active learning [46]. Ref. [48] suggested that the time period for the teaching learning process should be increased through the inclusion of technology essential for a blended learning program. Thus, an effective and quality-oriented blended learning course would recognize the requirements of teachers, address reservations and fears about changes and highlight the need for new strategies to be adopted for a better teaching learning process [45,49].

There are a number of professional development training courses that support teaching learning activities through blended learning [45,47]. However, according to Alvarado-Alcantar, Keeley and Sherrow [28], blended learning programs at a secondary level have been increasing since 2002, but the growth of the professional development of teachers with blended learning is still very low and it was needed to increase their technology accepted behavior that would be help for them to take advantages blended learning approach [50]. According to Archambault et al. [51], the level of teachers who previously attended courses for delivering online teaching was only 4.1%. Additionally, there is a smaller number of teachers available who were provided with professional training support to teach blended learning courses [51]. Moore, Robinson, Sheffield and Phillips [37] are of the opinion that when teachers are not supported with professional development training, they search for their own opportunities to learn. According to Refs. [46,52], less professional development training for teachers creates confusion regarding the effectiveness of the blended learning approach. The success of the blended learning approach for the professional development

of teachers to teach in a blended learning environment is related to well-planned and organized teacher training [17]. Limited research is available in comparison to higher education in the field of the professional development of teachers with a blended learning approach. There were only ten research works available on teacher education out of the 176 results of the meta-analysis of online research conducted from 1996 to 2008 [18]. It is also pertinent to mention that there is no empirical study available on the professional development of teachers with a blended learning approach [53]. As per the discussion thus far, it is worth mentioning that a blended learning approach is available, but limited research has been conducted on the professional development of teachers. Therefore, it was direly needed to investigate the effectiveness of blended learning empirically.

### 2.1. Evaluation of Training

Employee productivity is critical for businesses to succeed in the market and outperform their competition. Businesses' human resource departments or training departments can assist in achieving this goal by providing training opportunities for employees to increase productivity [54]. Employee knowledge, abilities and attitudes are developed through training in such a way that existing job performance is improved and employees are prepared to deal with future issues [55]. Employee performance is defined as the achievement of duties and assignments according to desired criteria [56]. During training, individuals learn the skills required to complete the assigned tasks effectively [57]. Therefore, from different perspectives, it is evident regarding the term training that a job description is provided to every individual working in an organization and they are supposed to perform their duties and tasks effectively; for this purpose, they are provided with training to achieve the desired tasks.

The ability to bring out the required results or outputs is referred to as effectiveness. It shows the attainment of the desired need. According to Homklin [58], the term training effectiveness is implied to improve the overall process of training in order to achieve established goals. Goals are established for any training program first, and then delivered to the target audience, and the achievement of set goals shows the effectiveness of that training program [59]. Thus, we can stated that the effectiveness of a training program can be determined by examining the training program with respect to the established goals. There are two basic concepts attached to the term training effectiveness, one is the training itself, while the other is the effects it triggered on the trainees [60]. Whatever is learnt from any executed training program can be implemented at the workplace to accomplish assigned tasks [61]. According to Mohammed Saad and Mat [62], an effective training process is the measurement of the process of delivering or executing the training program, while another element of training effectiveness is the improvement in the performance of individuals.

According to Desimone [59], the definition of a training evaluation is the well-organized gathering, analysis and synthesis of essential information required to determine a decision about the training effectiveness. The aforementioned definition clearly identifies detailed information about a training program, which can depict what is happening or has happened in any training program or development intervention. The evaluation of training programs is very important and useful for various stake holders, such as HRD professionals and managers to take appropriate measures regarding particular training programs and methods. Phillips [63] pointed out that there are also many benefits of training evaluations, including providing information about the achievement of objectives, identifying the strengths and weaknesses of a particular training program, highlighting the changes that are essential to improve training programs, identifying the individuals who can participate in future training programs, identifying the top and bottom performers of the development program, facilitating in marketing the evaluated program and providing a readily available database for the management to determine decisions. Zenger and Hargis [64] suggested the reasons for conducting a training evaluation, i.e., benefits for the budgets of training providers due to the authenticity of training as determined in the evaluation of a training program, because sometimes in budgets, funds are removed for human resource development due

to tough financial conditions. The second reason is it improves the credibility with top managers in an organization.

A training evaluation and training effectiveness are both theoretical approaches for measuring learning outcomes and training effectiveness. A training evaluation depicts a micro-view of training results, and training effectiveness puts emphasis on the learning as a whole and, consequently, provides the macro-view. The training evaluation identifies the learning benefits gained by the participants and improvements in their job performance. Effectiveness shows the benefits that an organization gained as a result of the development program. The results of the evaluation of training programs elaborate what happened as a result of the training intervention. The results of the effectiveness describe why those results happened and also provide recommendations to improve the training program [65].

According to Alvarez, Salas and Garofano [65], there are several methods for the evaluation of the effectiveness of training programs, but the most liked and acceptable method is the Kirkpatrick four-level evaluation model [66]. It provides a clear picture regarding the training program and what was and was not achieved [67].

### 2.2. Evaluation Models

The evaluation of programs and initiatives can be performed in a variety of ways. There were approximately 22 approaches gathered into five categories by [68]. These were categorized as pseudo-evaluations, quasi-evaluation studies, decision/accountability-oriented studies, improvement- and accountability-oriented evaluations and social agenda/advocacy-directed approaches. The various sub-categories are also available for the procedures as mentioned earlier. According to Ref. [69], there are four types of program evaluation methods. These include the program-oriented approach, consumer-oriented approach, decision-oriented approach and participant-oriented approach. According to Kashar [2], comprehensive judgements of a program's efficiency require expertise and a consumer-focused approach. The knowledge-based method is based on the technical competence of the program evaluator or a group of subject-matter experts. The main objective of both approaches is to find out the worth of the program and are applied in many fields. Kashar [2] argued that major empirical research has not utilized these approaches for program assessments.

The assessment methodologies are included in the program-oriented approach, which highlight a program's qualities. Logic models and the program theory are the focus of this approach. The logic model can be used to identify inputs, outputs and short-, medium- and long-term consequences. Logic models give logical relationships, simplified visual explaining and the underlying reasoning for a program or project [2].

The focus of evaluators is on the decision making in the decision-oriented approach [69]. Through the examination and identification of a program's decisions, success can be achieved in supporting managers in creating transparency and progress in a program [2,68]. Participants, sponsors, investors and program administrators, as well as those with a stake in the initiative, are targeted. One of this approach's strengths is its ability to improve the program members' awareness and utilization of program assessments [2].

Among the many evaluation models based on various techniques, there are Kirkpatrick's, CIPP, the experimental/quasi-experimental, the logic and Robert Stake's countenance models [24]. According to Royse, Thyer and Padgett [25], no model is of less importance, but can be best used as per the needs and requirements of researchers.

A researcher can analyze four components of a program's development and implementation using the CIPP program evaluation model, which includes the context, input, process and product [70]. The context is generally thought of as a type of needs assessment that aids in determining an organization's needs, assets and resources in order to develop programs that are appropriate for that organization. Evaluators work with stakeholders to identify program beneficiaries and create program goals in the context assessment component [71]. Various programs are investigated during the input evaluation component to see which ones most closely address the evaluated needs. The program's sufficiency, feasibility and

viability, as well as the financial, time and personnel expenses to the organization, must all be considered [71]. The process evaluation component focuses on the appropriateness and quality of the program's implementation. A program's intended and unexpected effects are identified via the product evaluation component [72].

### 2.3. Kirkpatrick Model for Training Evaluation

A training evaluation is very important for making training programs more effective in increasing the productivity of individuals. Effective training programs have positive effects on employees, enabling them to meet current challenges in their respective fields. Thus, training researchers have a consensus on the importance of training evaluations [73]. The Kirkpatrick model was used in this research for the training evaluation due to the following reasons:

Academics and human resource development practitioners utilize the Kirkpatrick four-level model to measure training success, since it is more comprehensible and well-established. Kirkpatrick updated this model multiple times, including in 1959, 1976, 1994, 1996 and 1998 [55]. According to [74], Donald Kirkpatrick's four-level assessment model is one of the most well-known and widely utilized evaluation models for the training and development of programs. The model provides a basic framework or vocabulary for talking about training outcomes and many sorts of data that can be used to assess how well training programs reached their objectives. Kashar [2] discussed that more than two hundred leadership training programs have been evaluated by using different levels of the Kirkpatrick model according to a research review provided by Throgmorton et al. [75]. This model provided a baseline for the emergence of other models of evaluation [2,55,58,76,77]. One or two Kirkpatrick levels are very often used by organizations to measure the effectiveness of training programs. The latest model of four levels is very comprehensive, simple and easy to use, which is why business and academic circles prefer to use it for training evaluations. According to Miller [78], the Kirkpatrick model provides an easy way of evaluating training programs. It is evident from various research works that finding out the training effectiveness by using the Kirkpatrick model has been accepted and well established [2,55,77,78]. This model also met their expectations. According to Homklin [58], the four levels of the Kirkpatrick model provide the most useful theoretical information about the training program under study. The major success of this model lies in the fact that it provides an opportunity to find changes in learner behaviors in the form knowledge and skills, and its application in the real working world [55]. Reio, Rocco, Smith and Chang [74] discussed that the Kirkpatrick model provides a well-organized and structured way for a formative and summative evaluation in such a way that the training process can be improved upon or completely reconstructed on the basis of the evaluation.

### 2.4. Criticism on Kirkpatrick Model

Though the four-level Kirkpatrick model has been greatly admired and well accepted there have still been criticisms. The first criticism is of its incompleteness and knowledge addition with the emergence of the four-level Kirkpatrick model training evaluation [55]. Guerci et al. [79] argued that a training evaluation is a dynamic process that is not as simple as that described by the Kirkpatrick model. He further argued that it is not possible to combine individual and company impacts in the assessment. Al-Mughairi [55] disagreed with the argument by saying that it is inappropriate to consider such considerations while using the Kirkpatrick model for a training evaluation. A meta-analytic review of the literature of the training effectiveness of the Kirkpatrick model was conducted by Alliger et al. [80], and it was found that the Kirkpatrick model provided a rough taxonomy of criteria and vocabulary. There was another finding at the same time that an over generalization and misunderstandings are possible due to the vocabulary and rough taxonomy for criteria [80]. The Kirkpatrick model also has the problem of being unclear on training. However, the model under discussion is much more popular and influential among practitioners [81]. According to Alliger, Tannenbaum, Bennett Jr., Traver

and Shotland [80], the Kirkpatrick model still communicates an understanding about training criteria. The model was also discussed in light of the assumptions present in the minds of trainers and researchers, including each successive level of the evaluation providing more comprehensive information to the organizations regarding the training effectiveness, every level being caused by the previous level and the levels being positively interconnected. It was also organized as a hierarchal training evaluation model due to the aforementioned assumptions [82].

According to Holton III [76], the involved construct is not properly defined in the four levels of the Kirkpatrick model, and, thus, simply provides a taxonomy of outcomes. Therefore, it is flawed as an evaluation model. He further argued that more tests are required to establish it as a completely defined and researchable evaluation model. Kirkpatrick [81] responded to this objection by saying that he was happy that his work was not considered as taxonomy, as trainers could not understand it completely. The aforementioned criticism brought a positive improvement in the model and, as a result, organizations still utilize this model on a considerable level.

### 2.5. Components of the Kirkpatrick Model and Their Relationships

The Kirkpatrick model mainly comprises of four levels of evaluation. These include reaction, learning, behavior and results. Following are the details of all four levels of this model:

### 2.5.1. Reaction Level

The first level deals with a participant's reaction towards various aspects of the training, including the trainer, training materials and management of training. Participants at this level express their feelings about the value of the training program as to whether it was productive or not [27]. The nature of this level is multi-dimensional, because of the fact that it measures trainee satisfaction about the overall aspects of the training program [83]. Researchers identified 11 categorizations of this level due to its multi-dimensional nature [27]. According to Reio, Rocco, Smith and Chang [74], the educational contents are the main concentration point of the level. If the participants give a positive reaction to level one, it means that, at this stage, training is marked as being good. However, this level is not a guarantee for the success of level two and further levels [78]. According to Miller [78], the Kirkpatrick model stresses the value of the level one assessment in various organizations. He further elaborated that a level one achievement is not a guarantee of success in learning, behavior or results of the organization [78]. According to Reio, Rocco, Smith and Chang [74], no behavioral improvement, which is level three, is very crucial to assess all responses of level one and level two. This level can be evaluated by using questionnaires [84]. The present study used the four-level Kirkpatrick model for the training evaluation to find out the effectiveness of the in-service training of vocational teachers.

### 2.5.2. Learning Level

This level deals with the assessment of the learning acquired by the participants in the form of knowledge, skills and attitude [85]. According to Miller [78], the learning level refers to the content achievement of training participants in the form of knowledge and skills as a result of participating in a training program. The second level of the model explains the success or failure of the training program, as it depicts the achievement of the set goals [78]. According to Kirkpatrick et al. [86], the second level determines what was gained by the training participants in the form of knowledge and acquired skills and competencies as a result of attending a training program. Level one and level two assessments are suggested by various researchers before moving onto level three [58]. The previous research literature showed that level two is the most common assessment level of training programs [78]. In order to demonstrate the value of a training program, it is very crucial to provide evidence of the knowledge and developed skills as a result of training [78]. The evaluation of learning is of utmost importance, because there can be no

change in behavior without learning [86]. Therefore, it is evident from the discussion above that an assessment of learning is very important to gauge the training effectiveness.

### 2.5.3. Behavior Level

This level assesses the application of the learned skills at the workplace [27]. Kirkpatrick and Hoffman [87] discussed that the behavior level measures how effectively the participants implement the knowledge and skills learned during the training program at their workplace. The assessment of the behavior level is very useful, because it shows the transfer of gained knowledge [74,78]. According to Reio, Rocco, Smith and Chang [74], without measuring the change in the behavior, it is not possible to measure the results of the training program. According to Kirkpatrick and Kirkpatrick, Fenton, Carpenter and Marder [86], as per his assumption, level three is not very appreciated. Trainers work really hard on level one and level two, because they have control over the aforementioned levels. For the implementation of new skills, sufficient time is required by the training participants [27].

### 2.5.4. Results Level

Level four provides the attainment of the results as a result of the training executed [58]. According to Miller [78], level four identifies the training effectiveness. The author of [2], Al-Mughairi [55] and the authors of [58,74,78,87] recognized level four as the most challenging and difficult to evaluate training effectiveness. According to Miller [78], Kirkpatrick was against the evaluation of only levels three and four while ignoring level one and two. According to Al-Mughairi [55,58,74,78], the role of all four levels of the Kirkpatrick model is very important in evaluating the effectiveness of a training program. The Kirkpatrick training evaluation model is implemented after the training has been delivered, so it is a summative-type evaluation that provides the overall effectiveness of the training program and guides whether to continue the training or if it needs some improvements [78]. According to Pineda-Herrero et al. [88], organizations invest a lot of resources into training and development, but hardly any of them evaluated it, whether the training served its purpose and whether it was worthwhile against the cost incurred on it.

The overall hypothesis of the study under discussion is,

**H0.** *There is a hierarchal relationship among the four levels of the Kilpatrick model for the training evaluation.*

**H0 (a).** *Level one influences level two.*

**H0 (b).** *Level two influences level three.*

**H0 (c).** *Level three influences level four.*

The overall conceptual framework of the study is given in Figure 1.

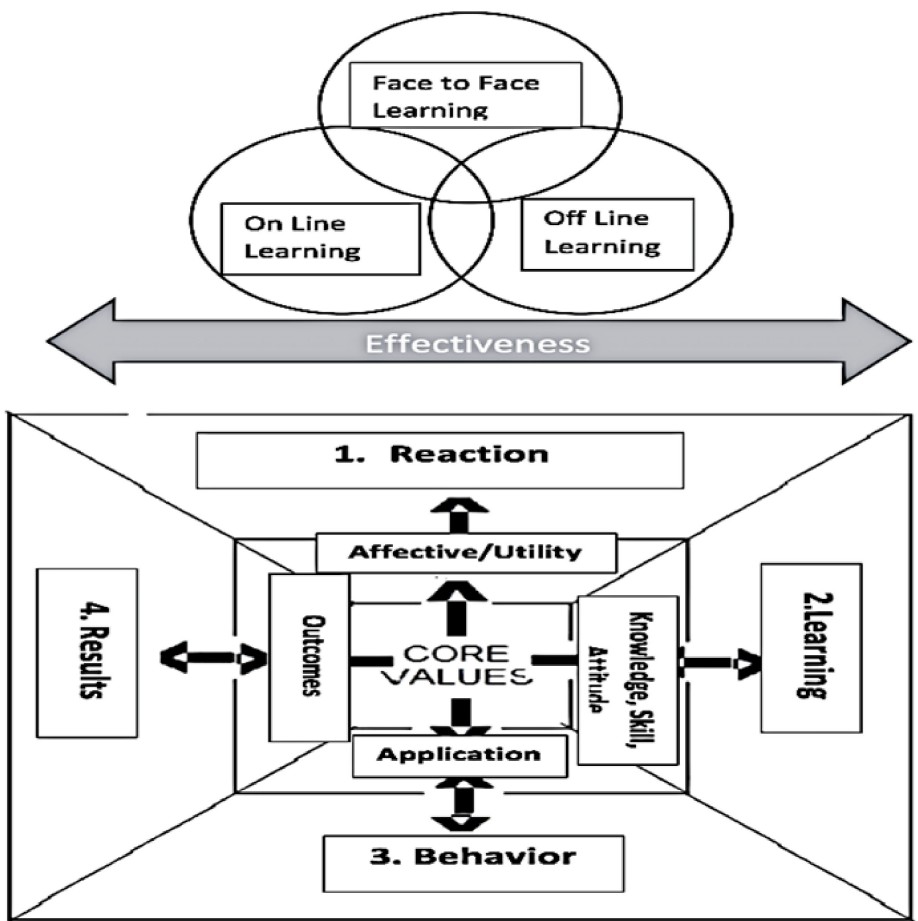

**Figure 1.** Theoretical model of blended learning and four levels of Kirkpatrick model.

## 3. Methodology

### 3.1. Research Design

The current study applied a survey design to collect data from participants. A survey was conducted on training participants by applying the Kirkpatrick model for the evaluation of the in-service teacher training program through blended learning approaches in Punjab. This study used the survey-based approach due to three reasons; first, a self-reported questionnaire would help to understand the behavior of the training participants; second, the statistical analysis of a big dataset would help in the generalization of the results for the population; third, the survey was collected on four levels of the Kirkpatrick model, making it comprehensive to understand the effectiveness of the training through various aspects.

### 3.2. Population and Sample

This study consisted of a population of trained in-service TVET teachers provided by GIZ under the TVET Reform Program at the institutes of the Punjab Vocational Training Council (PVTC). The total number of trained teachers was 781, and they were distributed across 204 vocational training institutes across Punjab affiliated with PVTC. A census method was used to survey all the training participants. The census method is used when the population size is small, thus, it involves the whole population to obtain voluminous information [89]. Therefore, in order to obtain the extensive amount of information, all the trained teachers were asked to fill out the training evaluation questionnaire. This was performed to attain more accurate and reliable results.

### 3.3. Training Evaluation Questionnaire (TEQ)

The training evaluation questionnaire (TEQ), developed by Homklin [58], was adopted and used with permission in the present study for the collection of quantitative data. It was translated to "Urdu" by an expert in order to facilitate the respondents of the sample of the study. The questionnaire was discussed with the language expert for translation. Certain changes in light of discussions with the supervisor were determined so that a clear understanding of the questionnaire would be provided for the respondents. Keeping in view the objectives and research questions of the current study, the survey questions were converted into a three-dimensional response, i.e., face-to-face, online and offline, as the training under study followed the blended learning approach. The agreed questionnaire was then sent to the experts for its validation on the scale of highly satisfied, satisfied and dissatisfied. The questionnaires were sent in two formats, through emails and hard forms. Certain changes were applied in the demographic portion, and questions were highlighted by the experts. The questionnaire was then validated by an expert adding those changes. The whole process took 1 to 2 months for the experts to validate the questionnaire. Pilot testing was run after the validation of the research questionnaire by four experts. This questionnaire had two parts.

First was the demographic part; age, qualification, experience and gender were added as demographics in this questionnaire.

Second, the TEQ was developed based on the Kirkpatrick model. This model had four levels. Each level was measured using a 5-point Likert-type scale, i.e., 1—strongly disagree; 2—disagree; 3—neutral; 4—agree; 5—strongly agree. Three modes of the blended learning approach, i.e., online, offline and face-to-face, were added in each question of the instrument.

Pilot testing was run after the validation of the research questionnaire by four experts.

### Data Collection Strategy

The data were collected from the trained in-service TVET teachers. For this purpose, we collected data through a large geographical area, area managers, principals and, in some cases, instructors were contacted to forward a link for filling out the questionnaire in their relevant area and institutes. The whole Punjab area was divided into three regions, central, north and south, headed by regional managers. Each region was further divided into areas, such as the central region, which was divided into four areas, whereas the north and south regions were divided into three areas each. Area managers directly dealt the affairs of the vocational training institutes of their respective areas. They all remained in contact throughout the process of data collection. Data were collected from the vocational training institutes through the aforementioned main employees of the organization, including in-service teachers. Teachers were kept informed as to ensuring the privacy of the provided information. Furthermore, if any reservations existed about the privacy, the information was removed.

### 3.4. Data Analysis Procedures

In this study, we applied both descriptive and inferential statistics. We applied analyses through SPSS and SmartPLS. The descriptive statistics were applied on demographics and measuring levels. The measurement model analysis and structural modeling analysis were applied through SmartPLS.

## 4. Results

### 4.1. Demographic Description

The total population of the study was 700 participants, for which we used a census sampling technique. The total number of useable questionnaires received by the researchers was 629; therefore, 629 participants took part in this training, out of which 69% were male and 31% were female. The age brackets of the respondents were 48.6% between 31 and 40 years, while 38.3% respondents were above 40 years and the remaining 13% respondents

were between 19 and 30 years. In total, 73.6% of respondents had more than 15 years' experience, while 18.6% of respondents had between 7 and 15 years' experience and the remaining 7.8% of respondents had from 1 to 6 years' experience. A total of 83% respondents did not have any special educational needs students in class, while only 17% of respondents had students with special educational needs in the class. In total, 80% of respondents were senior instructors and remaining the 20% of respondents were junior instructors (see Table 1).

**Table 1.** Participants' demographic information.

| Measure | Items | Frequency (n) | Percentage (%) |
|---|---|---|---|
| Gender | Male | 434 | 69.0 |
| | Female | 195 | 31.0 |
| | Total | 629 | 100.0 |
| Age Bracket | 30 to 40 Years | 306 | 48.6 |
| | Above 40 | 241 | 38.3 |
| | 19 to 30 years | 82 | 13.0 |
| | Total | 629 | 100.0 |
| Experience | More than 15 Years | 463 | 73.6 |
| | 7 to 15 years | 117 | 18.6 |
| | 1 to 6 | 49 | 7.8 |
| | Total | 629 | 100.0 |
| Inclusion of Students with Special Educational Needs in Classroom | No Students with Special Educational Needs | 522 | 83.0 |
| | Had Students with Special Educational Needs | 107 | 17.0 |
| | Total | 629 | 100.0 |
| Position Level | Senior Instructors | 503 | 80.0 |
| | Junior Instructors | 126 | 20.0 |
| | Total | 629 | 100.0 |

### 4.2. Structural Equation Modeling

The Smart-PLS version 3.2.8 was used for the data analysis in this research. A SEM analysis was comprised of a path analysis, confirmatory factor analysis and mediation analysis. A two-step procedure was used as recommended by [90], that comprised of the measurement of an outer model and the measurement of an inner model. A PLS-SEM was also considered an appropriate technique to measure the multi-variate analysis [91,92].

Appendix A shown the descriptive statistics, such as the mean and standard deviation.

### 4.3. Evaluation of Outer Model

The outer model measurement was based on the calculation of the construct reliability, single observed variables and the validity of the observed variables in relevance with the unobserved variables [93]. Single observed variables with an outer loading greater than 0.7 were acceptable [91], while items with values of less than 0.7 were discarded. The outer loading was arranged from 0.8 to 0.91, as shown in Table 2.

**Table 2.** Reliability and validity.

| | Œ | rho_A | CR | AVE |
|---|---|---|---|---|
| L1 | 0.978 | 0.979 | 0.979 | 0.673 |
| L2 | 0.946 | 0.948 | 0.956 | 0.756 |
| L3 | 0.945 | 0.946 | 0.956 | 0.783 |
| L4 | 0.949 | 0.954 | 0.957 | 0.716 |

Cronbach's alpha and the composite reliability (CR) measured the internal consistency of the construct reliability and showed that all latent construct values crossed the minimum threshold of the 0.7 level. The convergent validity of the construct was measured through the use of the average variance extracted (AVE) [94]. All constructs were found to be above 0.5 [91,95]. The results given in Table 2 show that the internal consistency and convergent validity were good.

The discriminant validity of the latent variables was measured at the next step. The discriminant validity distinguished the manifest variable from another construct in the path model, whereas the cross-loading values of the latent construct were higher than those of other constructs [96]. The discriminant validity was measured through the use of the HTMT ratio, as well as the cross-loading values [94]. The discriminant validity was proven satisfactory as all correlations were observed to be <1 [97], as shown in Table 3.

**Table 3.** Heterotrait–monotrait (HTMT) ratio of correlations.

|    | L1    | L2    | L3    |
|----|-------|-------|-------|
| L1 |       |       |       |
| L2 | 0.821 |       |       |
| L3 | 0.852 | 0.852 |       |
| L4 | 0.829 | 0.815 | 0.852 |

*4.4. Evaluation of Inner Model*

The measurement model was proved to be reliable and valid. The next step taken was to measure the inner model evaluation. It involved the observation of inner and out VIF values, predictive relevancy and relationships among constructs. The R-square as a coefficient of determination, t-statistic value, direct path coefficient (β value), effect size ($f^2$), goodness-of-fit (GoF) index and the predictive relevance of the model ($Q^2$) were the key points for measuring the structural model.

4.4.1. Coefficient of Determination $R^2$

The R-square measured the construct's overall effect size and variance explained in the structural model, which reflected the predictive accuracy of the model. In this model, L2's endogenous construct R-square value was 0.662, L3's R-square value was 0.73 and L4's construct R-square value was 0.67. According to [90,91], an R-square value greater than 0.75 is considered large, 0.5 is considered moderate and an R-square value of less than 0.26 is considered weak. All endogenous constructs were shown to have a larger predictivity in the given model, as shown in Table 4.

**Table 4.** Coefficient of determination $R^2$.

|    | R Square | R Square Adjusted |
|----|----------|-------------------|
| L2 | 0.789    | 0.789             |
| L3 | 0.81     | 0.809             |
| L4 | 0.816    | 0.816             |

4.4.2. Effect Size f-Square

The degree of the impact of the exogenous variables on the endogenous variables was measured with the f-square. The f-square values were considered as 0.02 (weak effect), 0.15 (moderate effect) and 0.35 (strong effect) [98]. L1 had a non-visible effect on L4, while L1 had the strongest effect on L2, with an f-square of 1.964, as shown in Table 5.

**Table 5.** f-square.

| | L2 | | L3 | | L4 | |
|---|---|---|---|---|---|---|
| | **Stats** | **Effect** | **Stats** | **Effect** | **Stats** | **Effect** |
| L1 | 3.741 | High | | | | |
| L2 | | | 4.254 | High | | |
| L3 | | | | | 4.445 | High |

### 4.4.3. Construct Cross-Validity Redundancy

Q-square statistics, or cross-validity redundancy, measure the quality of the path model. Blindfolding is usually used to measure the Q-square [99]. It shows how much a conceptual model can predict the endogenous constructs. The Q-square values in the SEM analysis had to be greater than zero. Figure 2 shows that all the endogenous constructs had Q-square values greater than zero, which was higher than the threshold level. This supported that the predictive relevance of the model was satisfactory for the endogenous constructs.

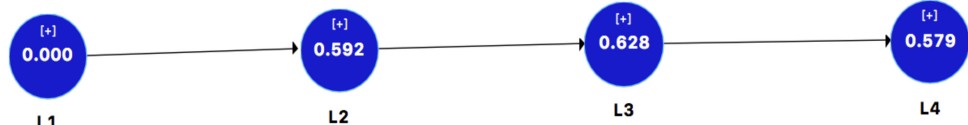

**Figure 2.** Constructs' cross-validity redundancy.

### 4.4.4. PLS Predict

The PLS predict was applied to measure the predictivity of the endogenous constructs. The $Q^2$_predict had to be greater than zero. Secondly, the MAE_LM values had to be greater than the MAE_PLS values. It was evident from the table that the $Q^2$_predict values were higher than zero and the MAE_LM values were higher than the MAE_PLS values, as shown in Table 6.

**Table 6.** PLS predict.

| | **RMSE_PLS** | **$Q^2$_Predict** | **RMSE_LM** | **RMSE (PLS-LM)** |
|---|---|---|---|---|
| L2.11 | 0.362 | 0.581 | 0.378 | −0.016 |
| L2.4 | 0.333 | 0.647 | 0.347 | −0.014 |
| L2.2 | 0.318 | 0.669 | 0.322 | −0.004 |
| L2.3 | 0.329 | 0.661 | 0.345 | −0.016 |
| L2.1 | 0.337 | 0.65 | 0.351 | −0.014 |
| L2.5 | 0.392 | 0.577 | 0.405 | −0.013 |
| L2.7 | 0.499 | 0.375 | 0.522 | −0.023 |
| L3.1 | 0.359 | 0.597 | 0.377 | −0.018 |
| L3.4 | 0.357 | 0.567 | 0.369 | −0.012 |
| L3.5 | 0.35 | 0.577 | 0.364 | −0.014 |
| L3.9 | 0.394 | 0.544 | 0.407 | −0.013 |
| L3.7 | 0.451 | 0.445 | 0.462 | −0.011 |
| L3.8 | 0.474 | 0.418 | 0.494 | −0.02 |
| L4.8 | 0.477 | 0.411 | 0.494 | −0.017 |
| L4.6 | 0.425 | 0.509 | 0.433 | −0.008 |
| L4.7 | 0.44 | 0.476 | 0.461 | −0.021 |
| L4.9 | 0.382 | 0.546 | 0.391 | −0.009 |
| L4.15 | 0.405 | 0.502 | 0.422 | −0.017 |
| L4.1 | 0.398 | 0.532 | 0.411 | −0.013 |
| L4.2 | 0.616 | 0.298 | 0.628 | −0.012 |
| L4.14 | 0.433 | 0.48 | 0.453 | −0.02 |
| L4.4 | 0.578 | 0.318 | 0.609 | −0.031 |

### 4.4.5. Goodness of Fit

An index that is applied to verify that a model effectively explains empirical data is known as the goodness-of-fit (GOF) index [99]. The value of the GoF index must be between 0 and 1, where 0.36 is a larger, 0.25 is a moderate and 0.10 is a smaller indication of the validation of a path model. A parsimonious and plausible model would show a good model fit index [100]. A good model fit can be measured with the following formula [99]:

$$\text{GoF} = \text{sqrt} \left( (\textit{average AVE}) \times (\textit{average } R^2) \right).$$

The calculation showed that the GoF value was 0.767, which reflected that the model was for empirical data and had a large predictivity power, as shown in Table 7.

**Table 7.** Goodness of fit index.

|         | **(AVE)** | **$R^2$** |
|---------|-----------|-----------|
| L1      | 0.673     |           |
| L2      | 0.756     | 0.789     |
| L3      | 0.783     | 0.809     |
| L4      | 0.716     | 0.816     |
| Average | 0.732     | 0.804     |
| GOF     |           | 0.767     |

The average of the standardized residual (SRMR) is an index between the hypothesis and observed covariance matrices [101]. The estimated fit of the model is measured through the use of the SRMR. A model with a SRMR $\leq 0.08$ is called a good fit [102]. Table 8 shows that the model had a SRMR = 0.039, whereas the Chi-square was equal to 4476.48 and the NFI value was 0.864, as shown in Table 8.

**Table 8.** Model fit summary.

|            | **Saturated Model** | **Estimated Model** |
|------------|---------------------|---------------------|
| SRMR       | 0.039               | 0.048               |
| d_ULS      | 1.608               | 2.345               |
| d_G        | 1.342               | 1.36                |
| Chi-Square | 4476.472            | 4507.728            |
| NFI        | 0.864               | 0.863               |

The VIF values were found below the threshold of five, as recommended by researchers. This showed that there was no multi-collinearity issues, as shown in Table 9.

**Table 9.** Inner VIF values.

|    | **L2** | **L3** | **L4** |
|----|--------|--------|--------|
| L1 | 1      |        |        |
| L2 |        | 1      |        |
| L3 |        |        | 1      |

## 5. Construct Measurements

Data were analyzed for the construct measurements. The calculation of the construct reliability, single observed variables and the validity of the observed variables was conducted in relation to the unobserved variables [93]. The single observed variables with an outer loading greater than 0.7 were acceptable [103], while items with values of less than 0.7 were discarded. The outer loading was arranged from 0.8 to 0.91, as shown in Table 10.

**Table 10.** Item loadings.

|        | L1    | L2    | L3    | L4    |
|--------|-------|-------|-------|-------|
| L1.1   | 0.836 |       |       |       |
| L1.10  | 0.814 |       |       |       |
| L1.11  | 0.743 |       |       |       |
| L1.12  | 0.72  |       |       |       |
| L1.13  | 0.868 |       |       |       |
| L1.14  | 0.822 |       |       |       |
| L1.15  | 0.853 |       |       |       |
| L1.16  | 0.795 |       |       |       |
| L1.17  | 0.835 |       |       |       |
| L1.18  | 0.858 |       |       |       |
| L1.19  | 0.859 |       |       |       |
| L1.2   | 0.848 |       |       |       |
| L1.20  | 0.861 |       |       |       |
| L1.22  | 0.833 |       |       |       |
| L1.23  | 0.794 |       |       |       |
| L1.24  | 0.782 |       |       |       |
| L1.25  | 0.747 |       |       |       |
| L1.3   | 0.827 |       |       |       |
| L1.4   | 0.839 |       |       |       |
| L1.5   | 0.827 |       |       |       |
| L1.7   | 0.86  |       |       |       |
| L1.9   | 0.806 |       |       |       |
| L1.6   | 0.825 |       |       |       |
| L2.1   |       | 0.858 |       |       |
| L2.11  |       | 0.862 |       |       |
| L2.2   |       | 0.9   |       |       |
| L2.3   |       | 0.911 |       |       |
| L2.4   |       | 0.917 |       |       |
| L2.5   |       | 0.865 |       |       |
| L2.7   |       | 0.763 |       |       |
| L3.1   |       |       | 0.909 |       |
| L3.4   |       |       | 0.896 |       |
| L3.5   |       |       | 0.901 |       |
| L3.7   |       |       | 0.873 |       |
| L3.8   |       |       | 0.837 |       |
| L3.9   |       |       | 0.893 |       |
| L4.1   |       |       |       | 0.862 |
| L4.14  |       |       |       | 0.854 |
| L4.15  |       |       |       | 0.894 |
| L4.2   |       |       |       | 0.707 |
| L4.4   |       |       |       | 0.706 |
| L4.6   |       |       |       | 0.903 |
| L4.7   |       |       |       | 0.888 |
| L4.8   |       |       |       | 0.867 |
| L4.9   |       |       |       | 0.901 |

Cronbach's alpha and the composite reliability (CR) measured the internal consistency of the construct reliability and showed that all latent construct values crossed the minimum threshold of the 0.7 level. The convergent validity of the construct was measured through the use of the average variance extracted (AVE) [94]. It was found to be above 0.5 for all constructs [95,104]. The results given in Table 10 show that the internal consistency and convergent validity were good.

## 6. Descriptive Statistics

Descriptive statistics were applied to find the satisfaction level of the training participants regarding the different levels of the Kirkpatrick model, i.e., reaction, learning, behavior and results. The table below clearly shows that the value of the mean scores

and standard deviation of reaction (M = 3.95; SD = 0.58) showed an amount of agreement, (M = 3.93; SD = 0.56) learning, (M = 3.90; SD = 0.56) behavior, (M = 3.97; SD = 0.56) and results (M = 3.97; SD = 0.58) showed the agreement of responses (See Table 11).

**Table 11.** Descriptive statistics of variables.

|  | N | Mean | Std. Deviation |
|---|---|---|---|
| L1 | 629 | 3.95 | 0.58 |
| L2 | 629 | 3.93 | 0.56 |
| L3 | 629 | 3.90 | 0.56 |
| L4 | 629 | 3.97 | 0.58 |

Note: level 1—reaction; level 2—learning; level 3—behavior; level 4—results.

*Direct and Specific Path Coefficients*

The $\beta$ coefficient in the regression analysis and the path coefficient in the PLs were considered to be the same. The hypothesis was tested through the $\beta$ values [105]. A unit variation in the independent variables effect on the dependent variables was denoted with $\beta$. The higher the $\beta$, the more substantial the effect of the independent variables on the dependent variables. The $\beta$ values were verified through t-statistics and significance level. The procedure of bootstrapping was used to measure the significance level of the hypothesis [106]. The results of the test of the path coefficient's significance and t-statistics at bootstrapping of 5000 subsamples are shown in Table 12. The direct effect of the reaction had a positive and significant relationship with learning ($\beta = 0.889$; t = 63.801; $p < 0.05$), therefore, Hypothesis H1 was accepted. The direct effect of learning had a positive and significant relationship with behavior ($\beta = 0.900$; t = 69.916; $p < 0.05$), therefore, Hypothesis H2 was accepted. The direct effect of behavior had a positive and significant relationship with the results ($\beta = 0.904$; t = 66.186; $p < 0.05$), therefore, Hypothesis H3 was accepted. The results of the specific indirect relationship among the reaction, learning, behavior and results clearly depicted that behavior was mediated through learning and the results ($\beta = 0.814$; t = 41.627; $p < 0.05$); thus, Hypothesis H4 was approved. Similarly, learning was mediated through the reaction and behavior ($\beta = 0.800$; t = 41.832; $p < 0.05$), and the reaction significantly influenced the results through learning and behavior ($\beta = 0.724$; t = 30.348; $p < 0.05$), therefore, Hypotheses H5 and H6 were accepted (See Table 12 and Figure 3).

**Table 12.** Direct and specific indirect effect.

| Hypotheses | Direct Relations | Coefficients | Mean | t-Stats | *p* Values | Results |
|---|---|---|---|---|---|---|
| H1 | L1　L2 | 0.889 |  | 63.801 | 0.000 | Accepted |
| H2 | L2　L3 | 0.900 |  | 69.916 | 0.000 | Accepted |
| H3 | L3　L4 | 0.904 |  | 66.186 | 0.000 | Accepted |
|  |  | Indirect Relations |  |  |  |  |
| H4 | L2　L3　L4 | 0.814 |  | 41.627 | 0.000 | Accepted |
| H5 | L1　L2　L3 | 0.800 |  | 41.832 | 0.000 | Accepted |
| H6 | L1　L2　L3　L4 | 0.724 |  | 30.348 | 0.000 | Accepted |

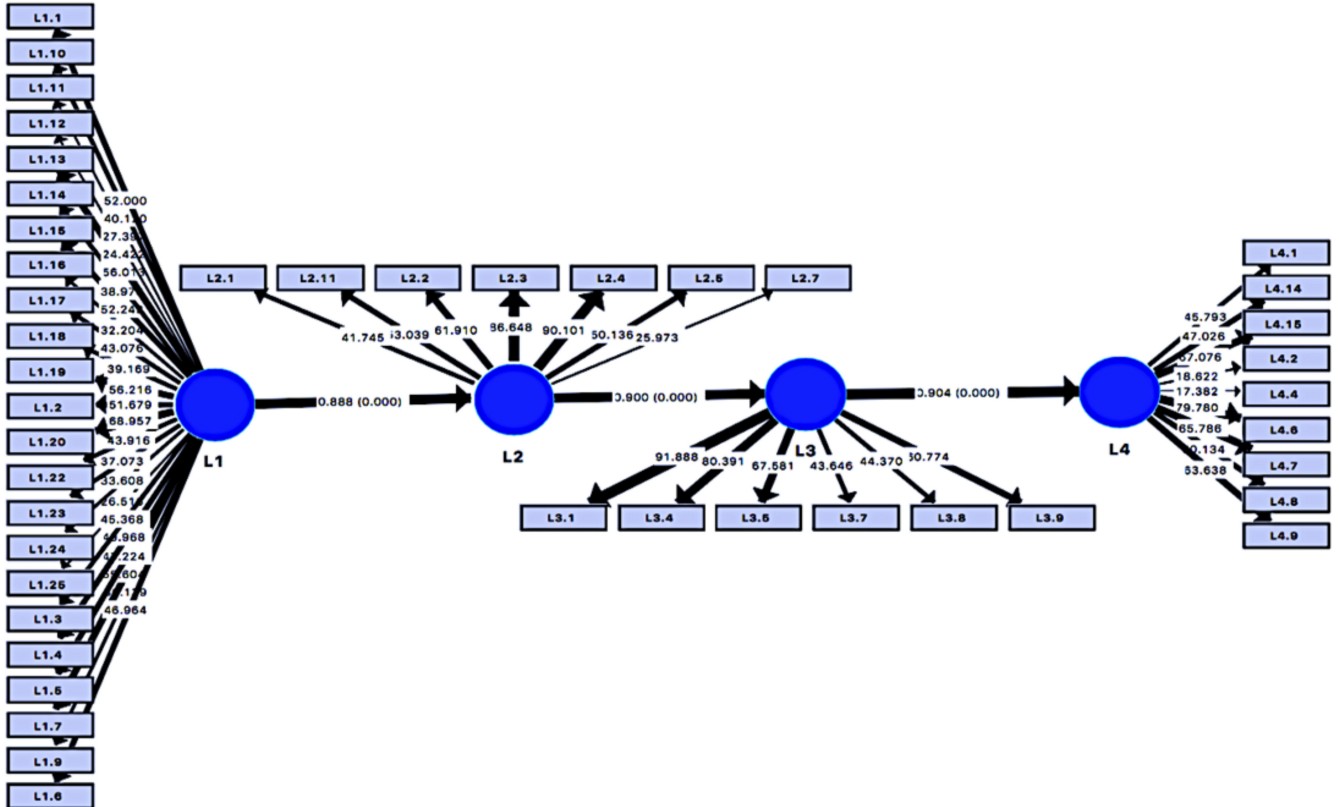

**Figure 3.** Relationships of variables.

## 7. Discussion

The present study aimed at evaluating the effectiveness of in-service vocational education teacher training through blended learning in Punjab. A training evaluation was conducted by using the four levels of the Kirkpatrick model. Further, the aim was to find the hierarchal relationships among the four levels of the Kirkpatrick model, and to comparative the effectiveness of the various modes of blended learning, i.e., online, offline and face-to-face. To the best of our knowledge, this was the first study in a developing country to evaluate the available in-service vocational teacher training programs through blended learning approaches. In this section, we discussed the knowledge contributions and practical implications of the study.

The Kirkpatrick training evaluation model was used to evaluate the training effectiveness of in-service vocational teacher training with blended learning approaches in Punjab. It was evident from the findings that respondents agreed on the effect of level one, the reaction, which meant satisfaction from the trainer, training materials, training environment and all modes of blended learning approaches, i.e., online, face-to-face and offline. Bates [107] and Kirkpatrick, Fenton, Carpenter and Marder [86] showed that the reaction level is very important in evaluating a training program. Results showed that, for the second level of the Kirkpatrick model, the in-service vocational teachers showed their consensus on gaining knowledge, skills and work-related attitudes through the various modes of training with blended learning. The results indicated that vocational teachers learned different learning approaches, teaching methods, activity-based learning, the division of training contents into sections to make content understandable and easy for students to understand, classroom management, time management, making lessons interesting, creating attractive PowerPoint presentations, lesson planning, working at the workshop floor level and the use of IT resources. The results for the behavior level, which is the third level of the Kirkpatrick model, revealed that participants also agreed that they implemented the learned skills at their workplace, which benefited them and their students. Finally, the respondents were also satisfied with the results level that reflected the achievement

of training objectives. The results of the current study confirmed the outcomes of the previous studies concerning the training effectiveness of in-service vocational teachers with blended learning approaches [108,109]. According to Mir [110], the satisfaction level is very important, and predicts the successful achievement of the learning outcomes. Kashar [2] argued that teacher training brings improvements into the teaching practices of the teachers, which resulted in the better learning of their students. Puangrimaggalatung [111] viewed teacher training programs as beneficial for their professional development in terms of improvements in the teaching process and practices. According to Jahari [112], the teacher training improved the quality of teachers.

Four levels of the Kirkpatrick model were used in this study to find the hierarchal relationship for in-service vocational teacher training through blended learning approaches in Punjab. The results showed that all observed variables had sufficient correlations. The reaction level had a strong relationship with the knowledge level, the knowledge level had a strong relationship with the behavior level and the behavior level had a strong relationship with the results level. However, the results level and reaction level had a weak relationship. The hierarchal relationship was also measured in various other empirical research works in the past, such as [2,41,58,80,110,113–120]. The hierarchal relation among the levels of the Kirkpatrick model was also extracted by various researchers in their research studies, e.g., [58,61,121–128] and meta-analyses of training evaluation studies by [80,113]. The current research endorsed the results of previous research works in the vocational sector.

## 8. Conclusions

The study conducted an evaluation of in-service vocational teacher training programs in the context of a blend of face-to-face, online and offline learning approaches. The study concluded that respondents agreed as to the training effect of level one of the Kirkpatrick model, reaction, which meant satisfaction from the trainer, training materials, training environment and all modes of blended learning approaches, i.e., online, face-to-face and offline also worked well. Moreover, it was concluded that training influenced the second level of the Kirkpatrick model for the in-service vocational teachers, which including gaining knowledge, skills and work-related attitude through various modes of training with blended learning. It was also concluded that the vocational teachers learned different learning approaches, teaching methods, activity-based learning, the division of training content into sections to make it more understandable and easy for students, classroom management, time management, making lesson interesting, creating attractive PowerPoint presentations, lesson planning, working at the workshop floor level and the use of IT resources. We also concluded that the third level of the Kirkpatrick model training had effects on the behavior dimension, that the participants also agreed that they implemented the learned skills at their workplace, which benefited them and their students. It was also concluded that at the satisfaction level, the participants reflects the achievement of training objectives.

Vocational training institutes (VTIs) should provide at least one desktop with an internet connection in the lab of every trade being taught at that institute. Different trades are taught at vocational centers, e.g., auto-mechanics, dress making, food and cooking, beauticians, computer operators, auto CAD, auto electricians, etc. This could help teachers of non-computer trades to complete their assigned tasks during a non-presence phase and can also implement the learned skills in real work environments. It would increase the effectiveness of in-service teacher training with the blended learning approach. For this purpose, VTIs should allocate a budget for IT resources in non-computer trades. Like some VTIs, the heads of other institutes should also nominate IT instructors to provide guidance and support to the non-computer trade instructors. In order to implement this strategy, the head office should send instructions to the heads of vocational institutes to provide such technical support to the instructional staff participating in the in-service training with the blended learning approach. In addition to this support, basic IT skills training can be provided to non-IT trade instructors from their IT instructors prior to nomination for

in-service training through blended learning approaches. For this purpose, every institute could use its existing lab after the institute's working hours. This would help to obtain the maximum benefits of the blended learning approaches.

Blended learning is an innovative approach that provides flexible learning opportunities in different learning environments. It is a new approach for the vocational sector of Pakistan, so future research recommendations are as follows: The present research was conducted to evaluate blended learning for the vocational stream, and now, it is recommended to conduct research in the technical stream comprising teaching at a diploma level. A comparative analysis research of blended learning effectiveness for different occupational trades, e.g., fashion design, computer operators, electricians, clinical assistants, etc., can be conducted.

**Author Contributions:** Conceptualization, M.Z.A., Y.W. and P.S.-H.; methodology, M.Z.A. and P.S.-H.; software, M.Z.A., Y.W., J.I. and P.S.-H.; validation, M.N.A., Y.W., J.I. and P.S.-H.; formal analysis, M.Z.A., Y.W., J.I. and P.S.-H.; investigation, M.Z.A., Y.W., J.I. and P.S.-H.; resources, M.N.A., Y.W., J.I. and P.S.-H.; data curation, M.N.A., Y.W., J.I. and P.S.-H.; writing—original draft preparation, M.Z.A., Y.W., J.I. and P.S.-H.; writing—review and editing, M.Z.A., Y.W., J.I. and P.S.-H.; visualization, M.Z.A., Y.W., J.I. and P.S.-H.; supervision, M.Z.A., Y.W., J.I. and P.S.-H.; project administration, M.N.A., Y.W., J.I. and P.S.-H.; funding acquisition, M.Z.A., Y.W., J.I. and P.S.-H. All authors have read and agreed to the published version of the manuscript.

**Funding:** Open access funding provided by University of Helsinki.

**Institutional Review Board Statement:** The data collected in the present study were conducted after the approval of the board of advance study at the University of Management and Technology, Lahore, Pakistan, dated June 2019. We confirm that all methods used in this study were carried out in accordance with relevant guidelines and regulations.

**Informed Consent Statement:** The participation of students was completely voluntary and informed consent was obtained from all participants or, if participants were under 18, from a parent and/or legal guardian. Consent for publication was not applicable.

**Data Availability Statement:** The datasets used and/or analyzed during the current study are available from the corresponding author upon reasonable request.

**Conflicts of Interest:** The authors declare no conflict of interest.

## Appendix A

Questionnaire

Blended Learning: Blended learning is a combination of online, offline and face to face learning.

Online Learning: Online learning means learning through internet, email and mobile phones.

Face to Face Learning: It means classroom learning through lectures, presentations, group work and discussion, etc.

Offline Learning: Offline learning means self-learning by using the educational material in the form of training manual and library provided in CD-ROM and competing assigned tasks.

Scales 1 2 3 4 5 = Very dissatisfied, Dissatisfied, Neither, Satisfied, Very Satisfied.

| Items | Mode of Training | Strongly Dissatisfied | Dissatisfied | Neither | Satisfied | Strongly Satisfied |
|---|---|---|---|---|---|---|
| **Part 1: Level 1: Reaction** | | | | | | |
| How satisfied are you with the instructor's knowledge of training material and subject matter? | Online | | | | | |
| | Face-to-face | | | | | |
| | Offline | | | | | |

| Items | Mode of Training | Strongly Dissatisfied | Dissatisfied | Neither | Satisfied | Strongly Satisfied |
|---|---|---|---|---|---|---|
| How satisfied are you with the instructor's ability to make you keep interest in training sessions? | Online | | | | | |
| | Face-to-face | | | | | |
| | Offline | | | | | |
| How satisfied are you with the instructor's presentation and explanation of training materials? | Online | | | | | |
| | Face-to-face | | | | | |
| | Offline | | | | | |
| How satisfied are you with the instructor's responsiveness to trainee questions and problems? | Online | | | | | |
| | Face-to-face | | | | | |
| | Offline | | | | | |
| How satisfied are you with instructor's ability to have good relationships to you individually? | Online | | | | | |
| | Face-to-face | | | | | |
| | Offline | | | | | |
| How satisfied are you with instructor's overall effectiveness? | Online | | | | | |
| | Face-to-face | | | | | |
| | Offline | | | | | |
| How satisfied are you with the availability of training courses for teachers. | Online | | | | | |
| | Face-to-face | | | | | |
| | Offline | | | | | |
| How satisfied are you with the communication of training information to trainees? | Online | | | | | |
| | Face-to-face | | | | | |
| | Offline | | | | | |
| How satisfied are you with the quality of training services provided to trainee? | Online | | | | | |
| | Face-to-face | | | | | |
| | Offline | | | | | |
| How satisfied are you with registration process and information you received prior to training? | Online | | | | | |
| | Face-to-face | | | | | |
| | Offline | | | | | |
| How satisfied are you with quality of training provided by instructors? | Online | | | | | |
| | Face-to-face | | | | | |
| | Offline | | | | | |
| How satisfied are you with the fairness of the course exam? | Online | | | | | |
| | Face-to-face | | | | | |
| | Offline | | | | | |
| How satisfied are you with coverage and importance of material tested? | Online | | | | | |
| | Face-to-face | | | | | |
| | Offline | | | | | |
| How satisfied are you with feedback you received as result of course testing? | Online | | | | | |
| | Face-to-face | | | | | |
| | Offline | | | | | |

| Items | Mode of Training | Strongly Dissatisfied | Dissatisfied | Neither | Satisfied | Strongly Satisfied |
|---|---|---|---|---|---|---|
| How satisfied are you with communication of training's objectives in clear, understandable terms? | Online | | | | | |
| | Face-to-face | | | | | |
| | Offline | | | | | |
| How satisfied are you with match of training's objectives with your idea of what would be taught? | Online | | | | | |
| | Face-to-face | | | | | |
| | Offline | | | | | |
| How satisfied are you with training's emphasis on most important information? | Online | | | | | |
| | Face-to-face | | | | | |
| | Offline | | | | | |
| How satisfied are you with quality of this training overall? | Online | | | | | |
| | Face-to-face | | | | | |
| | Offline | | | | | |
| How satisfied are you with the length of training session? | Online | | | | | |
| | Face-to-face | | | | | |
| | Offline | | | | | |
| How satisfied are you with the quality of training materials? | Online | | | | | |
| | Face-to-face | | | | | |
| | Offline | | | | | |
| How satisfied are you with the audio and visual aids used by the instructor? | Online | | | | | |
| | Face-to-face | | | | | |
| | Offline | | | | | |
| How satisfied are you with the supplies and equipment for this training? | Online | | | | | |
| | Face-to-face | | | | | |
| | Offline | | | | | |
| How satisfied are you with classrooms, furniture, learning environment, etc.? | Online | | | | | |
| | Face-to-face | | | | | |
| | Offline | | | | | |
| **Level 2: Learning** | | | | | | |
| My knowledge increased as a result of this training. | Online | | | | | |
| | Face-to-face | | | | | |
| | Offline | | | | | |
| I feel that newly learned knowledge can do my current job better. | Online | | | | | |
| | Face-to-face | | | | | |
| | Offline | | | | | |
| I could improve my knowledge to find out problems in the daily job. | Online | | | | | |
| | Face-to-face | | | | | |
| | Offline | | | | | |
| | Offline | | | | | |

| Items | Mode of Training | Strongly Dissatisfied | Dissatisfied | Neither | Satisfied | Strongly Satisfied |
|---|---|---|---|---|---|---|
| I remember almost every knowledge covered in the training. | Online | | | | | |
| | Face-to-face | | | | | |
| | Offline | | | | | |
| My skills are increased as a result of this training. | Online | | | | | |
| | Face-to-face | | | | | |
| | Offline | | | | | |
| I feel that my newly learned skill can do current job better. | Online | | | | | |
| | Face-to-face | | | | | |
| | Offline | | | | | |
| I could improve my skill to find out problems in the daily job. | Online | | | | | |
| | Face-to-face | | | | | |
| | Offline | | | | | |
| **Level 3: Application & Implementation** | | | | | | |
| Using the new knowledge and skills has helped me to improve my work. | Online | | | | | |
| | Face-to-face | | | | | |
| | Offline | | | | | |
| I can accomplish job tasks better by using new knowledge and skills | Online | | | | | |
| | Face-to-face | | | | | |
| | Offline | | | | | |
| I make fewer mistakes in production when using new knowledge and skills | Online | | | | | |
| | Face-to-face | | | | | |
| | Offline | | | | | |
| I remember the main topics learned in the training. | Online | | | | | |
| | Face-to-face | | | | | |
| | Offline | | | | | |
| I easily say several things learned in the training. | Online | | | | | |
| | Face-to-face | | | | | |
| | Offline | | | | | |
| Never thought again about the training content (-). | Online | | | | | |
| | Face-to-face | | | | | |
| | Offline | | | | | |
| **Level 4: Individual and institutional Results** | | | | | | |
| This training was a worthwhile investment in my career development. | Online | | | | | |
| | Face-to-face | | | | | |
| | Offline | | | | | |
| This training has helped prepare me for other job opportunities within the other organizations. | Online | | | | | |
| | Face-to-face | | | | | |
| | Offline | | | | | |

| Items | Mode of Training | Strongly Dissatisfied | Dissatisfied | Neither | Satisfied | Strongly Satisfied |
|---|---|---|---|---|---|---|
| I am seeking for more chances to change job by using this training. | Online | | | | | |
| | Face-to-face | | | | | |
| | Offline | | | | | |
| I have been given verbal praise for applying new knowledge and skills. | Online | | | | | |
| | Face-to-face | | | | | |
| | Offline | | | | | |
| The training improved my job involvement | Online | | | | | |
| | Face-to-face | | | | | |
| | Offline | | | | | |
| This training has made me feel more committed to my institutions. | Online | | | | | |
| | Face-to-face | | | | | |
| | Offline | | | | | |
| This training has given me a sense of loyalty to my institution. | Online | | | | | |
| | Face-to-face | | | | | |
| | Offline | | | | | |
| This training was worthwhile investment for my institution. | Online | | | | | |
| | Face-to-face | | | | | |
| | Offline | | | | | |
| This training has made me feel like I will stay with my institution for many years. | Online | | | | | |
| | Face-to-face | | | | | |
| | Offline | | | | | |

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
