# Peer review of "Evaluation of In-Service Vocational Teacher Training Program: A Blend of Face-to-Face, Online and Offline Learning Approaches"

_sustainability, doi:10.3390/su142113906_

Round 1

Reviewer 1 Report

I have the following recommendations:

1. The title should be rephrased to be simpler and clearer. Now, it is not very clear and some corrections should be made regarding the English

2. Avoid abbreviations in the abstract. Shorten your phrases in the abstract, they are too long and create confusion especially because you are not an English native. Line 28, I would advise the deletion of the word would: the results are useful.... you have to be convinced. 

3. Line 35 is not a paragraph, so connect the phrase with other phrases to make sense. 

4. The paper needs to be checked by a professional in English, they are many small mistakes. 

5. Figure 1 should be made clearer. The text is not seen very well. Also figure 3

6. The 4-th heading is also about methodology. Try to delimitate in your paper what is method, what is about results because they are mixed-up. Smart-pls should be described as a method in Methodology and all results regarding the findings, the sample etc should be in Results. 

7. Line 557. Is there stamp or step? I am familiar with the method you used but do not understand what stamp is??

8. Line 561: as well as cross loading values was measured, Please correct the English

9. Line 563, Please write lesser than 1 not lesser <1

10. Discussion and conclusions should be two headings. In the Discussion you present other papers that confirm or not your findings and conclusions should present the theoretical and practical implications, limitations of your research and future research directions.

11. Instead of future researches, use future research directions. 

12. You have various styles of formatting through the text. try to be consistent

13. In the Discussion, you should present the papers of author researchers confirming or not your research. Try to add also a few references from 2021-2022. 

14. References are not formatted in accordance with the journal guidelines, but I suppose they will inform you about this aspect. 

Author Response

Reviewer 1:

Comment 1. The title should be rephrased to be simpler and clearer. Now, it is not very clear and some corrections should be made regarding the English.

Answer: We have revised title of the article as per the recommendation of the reviewer.

Comment 2. Avoid abbreviations in the abstract. Shorten your phrases in the abstract, they are too long and create confusion especially because you are not an English native. Line 28, I would advise the deletion of the word would: the results are useful.... you have to be convinced. 

Answer: We have removed abbreviation from the abstract as per the recommendation of the reviewer. We have deleted the word “would”. We have changed the sentence “the results are useful.... you have to be convinced” in abstract.

Comment 3. Line 35 is not a paragraph, so connect the phrase with other phrases to make sense. 

Answer: We have deleted this line because the next sentence has that information as a opening sentence.

Comment 4. The paper needs to be checked by a professional editor, they are many small mistakes. 

Answer: We have get checked by a native English language expert, and corrected small mistakes.

Comment 5. Figure 1 should be made clearer. The text is not seen very well. Also figure 3

Answer: We tried to make Figure 1 & 3 more clear. They have improved. Kindly check it now.

Comment 6. The 4-th heading is also about methodology. Try to delimitate in your paper what is method, what is about results because they are mixed-up. Smart-pls should be described as a method in Methodology and all results regarding the findings, the sample etc should be in Results. 

Answer: We changed the heading and corrected its contents too. Kindly check it. We delimited our method too. Moreover, we have incorporated all changes recommended by the reviewers, kindly check it.

Comment 7. Line 557. Is there stamp or step? I am familiar with the method you used but do not understand what stamp is??

Answer: It is word step.

Comment 8. Line 561: as well as cross loading values was measured, Please correct the English

Answer: We corrected the language.

Comment 9. Line 563, Please write lesser than 1 not lesser <1

Answer: We have corrected and used standard interpretation language in our manuscript.

Comment 10. Discussion and conclusions should be two headings. In the Discussion you present other papers that confirm or not your findings and conclusions should present the theoretical and practical implications, limitations of your research and future research directions.

Answer: We have divided discussion and conclusions into two headings. In the discussion, we presented other papers which confirmed our findings. Conclusions presented the theoretical and practical implications, limitations of research and future research directions. We incorporated all suggested changes. Kindly check them in manuscript.

Comment 11. Instead of future researches, use future research directions. 

Answer: We corrected term as “future research directions”. Kindly check it from manuscript.

Comment 12. You have various styles of formatting through the text. try to be consistent

Answer: We changed the styles according to the journal requirement.

Comment 13. In the Discussion, you should present the papers of author researchers confirming or not your research. Try to add also a few references from 2021-2022. 

Answer: We have used some research from 2021-2022. 

Comment 14. References are not formatted in accordance with the journal guidelines, but I suppose they will inform you about this aspect.

Answer: We corrected the references’ formatted in accordance with the journal guidelines.

Reviewer 2 Report

Generally, this article was qualified since it brought almost complete data processing. However, several items should be followed to encourage its quality:

1. Explain the population and sample to obtain representativeness. Please present the respondent's demography in table to enhance its readability.

2. Please deliver list of question/instrument used by this research. This article did not reveal what were filled by respondents, but suddenly portray he statistics of processed data.

3. Highlight much more problems occurred in Introduction, especially gap between expectation and reality. Furthermore, please justify whether this research solved the problem or not.

Thank you

Author Response

Comment 1. Explain the population and sample to obtain representativeness. Please present the respondent's demography in table to enhance its readability.

Answer: We have explained the population and representativeness of sample. Moreover, we inserted Table related to respondent's demography.

Comment 2. Please deliver list of question/instrument used by this research. This article did not reveal what were filled by respondents, but suddenly portray he statistics of processed data.

Answer: We have delivered list of question/instrument used in this research. We requested the editor appendix 1 is only for raw data. Kindly do not publish it. We have explained respondents in data collected and demographic analysis part. Kindly check it.

Comment 3. Highlight much more problems occurred in Introduction, especially gap between expectation and reality. Furthermore, please justify whether this research solved the problem or not.

Answer: We have highlighted more problems such as gaps between reality. We have provided the details how this study resolve the problem in introduction part.

Reviewer 3 Report

This work about "Measuring the effectiveness of in-service vocational education 2 teachers’ training program through blended learning approaches in Pakistan" is a well organized and sound research but the outcome could be much better and should be improved

1) First of all there is correlation of this work with a recent work by the majority of authors "Analyzing an Appropriate Blend of Face-to-Face, Offline and Online Learning Approaches for the In-Service Vocational Teacher’s Training Program", published in Int. J. Environ. Res. Public Health 2022, 19, 10668, MDPI, 26/8/2022.  This paper is not cited in the new paper although there is some correlation. Not exactly overlap since here the theoretical framework is the Kirkpatrick model. However , there is connection and this recent work should be cited defining the context of the new one. 

2) The most important aspect, however, to be improved is that the authors have analyzed statistically only with respect to L1, L2, L3, L4 level variables of the Kirkpatrick model the 629 samples questionnaire.  The statistical analysis is thorough. But it is done in the whole population.  We don't get any information on what happens in separate groups. For instance , online, offline and face to face groups , if there are such specific groups. It is not clear if each participant had , online, offline and face to face learning integrated.  If this is the case please clarify it. Then, most important grouping for analyzing again Kirkpatrick model is rural areas / cities. What happens there with L1, L2, L3, L4 analysis. Also, men,. women grouping. Are there any differences between these kind of groupings??

You can improve a lot your paper giving us much more information on Kirkpatrick model application and effects in different groupings of your sample with regards to blended training of teachers.

Author Response

Comment 1. First of all there is correlation of this work with a recent work by the majority of authors "Analyzing an Appropriate Blend of Face-to-Face, Offline and Online Learning Approaches for the In-Service Vocational Teacher’s Training Program", published in Int. J. Environ. Res. Public Health 2022, 19, 10668, MDPI, 26/8/2022.  This paper is not cited in the new paper although there is some correlation. Not exactly overlap since here the theoretical framework is the Kirkpatrick model. However , there is connection and this recent work should be cited defining the context of the new one. 

Answer: We have cited study which reviewer mentioned in the comments.

Comment 2. The most important aspect, however, to be improved is that the authors have analyzed statistically only with respect to L1, L2, L3, L4 level variables of the Kirkpatrick model the 629 samples questionnaire.  The statistical analysis is thorough. But it is done in the whole population.  We don't get any information on what happens in separate groups. For instance , online, offline and face to face groups , if there are such specific groups. It is not clear if each participant had , online, offline and face to face learning integrated.  If this is the case please clarify it. Then, most important grouping for analyzing again Kirkpatrick model is rural areas / cities. What happens there with L1, L2, L3, L4 analysis. Also, men,. women grouping. Are there any differences between these kind of groupings??

Answer: We applied basic screening tests to proceed for groups comparison based on gender. But basic tests did not fulfill the criteria to split population into two groups (make and female) ti proceed for Multiple Group Analysis (MGA) in SmartPLS. therefore we did not apply group difference tests. Moreover, this study’ explored a blended approach and it meets the scope.

Comment 3. You can improve a lot your paper giving us much more information on Kirkpatrick model application and effects in different groupings of your sample with regards to blended training of teachers.

Answer: Since our scope of the study was limited to the evaluation of blended learning approaches for in-service vocational teacher training. There are papers available on f-To-f, online and offline groups comparisons but it was needed to evaluate overall blended learning. Therefore, we focused on the evaluating overall training program using four levels of the Kirkpatrick model.

Round 2

Reviewer 1 Report

I am satisfied with the changes except one. Maybe I was not clear enough. I suggest the authors to incorporate Headings 7 and 8 into 6 - Conclusions. Practical implications and Limits and future research directions should be part of the conclusion. It is no need to put them as subheadings, just paragraphs in the Conclusion section. I think your paper will look better this way. 

Great success in your career and research.
